# Open-Ended Reinforcement Learning with Neural Reward Functions

**Robert Meier** *
Department of Computer Science
ETH Zürich
Zürich, Switzerland
romeier@inf.ethz.ch

**Asier Mujika** *
Department of Computer Science
ETH Zürich
Zürich, Switzerland
asierm@inf.ethz.ch

## Abstract

Inspired by the success of unsupervised learning in Computer Vision and Natural Language Processing, the reinforcement learning community has recently started to focus more on unsupervised discovery of skills. Most current approaches, like DIAYN or DADS, optimize some form of mutual information objective. We propose a different approach that uses reward functions encoded by neural networks. These are trained iteratively to reward more complex behavior. In high-dimensional robotic environments our approach learns a wide range of interesting skills including front-flips for HALF-CHEETAH and one-legged running for HUMANOID. It is the first skill discovery algorithm that can learn such skills without relying on any form of feature engineering. In the pixel-based Montezuma's Revenge environment our method also works with minimal changes and it learns complex skills that involve interacting with items and visiting diverse locations. The implementation of our approach can be found here.

## 1 Introduction

Deep reinforcement learning (RL) has proven to be very successful in many challenging tasks (Mnih et al., 2015; Silver et al., 2017b; Berner et al., 2019). These were considered intractable just a few years ago. However, current methods require enormous amounts of compute to achieve great performance in individual tasks. Most of the time, previous models can not be utilised when new tasks are considered, even if the environment does not change. This was also the case in Computer Vision and Natural Language Processing. However, recently unsupervised learning has been shown to be very effective in both fields. By using task agnostic pre-training schemes (Tenney et al., 2019; Brown et al., 2020; Caron et al., 2021), unsupervised models can solve most tasks with minimal or even no fine-tuning. Unsupervised reinforcement learning aims to bring similar successes to the reinforcement learning community.

Agents which learn a wide range of unsupervised skills may be able to leverage those to solve new tasks faster and with minimal fine-tuning.

Here, we propose a new method for open-ended, unsupervised skill discovery.

We devise an iterative process which creates pairs of neural reward functions and policies. The policy optimizes the corresponding reward function. Each of them corresponds to a skill that the agent has learned. The neural reward function is a neural network that maps the current observation to a scalar reward. In each iteration, the neural reward function is modified to differ from the previous one. This results in increasing the complexity of the encoded task. Then, a new skill is trained to optimize this reward function. We devise several techniques to transfer the knowledge from previously learned

---

*equal contribution

36th Conference on Neural Information Processing Systems (NeurIPS 2022).

skills. These mechanisms enable learning the most complex reward functions that our method creates. In fact, we show that some of the functions are impossible to learn from scratch.

We empirically test our framework in a diverse set of environments. First, we apply it to a simple 2d navigation task. This lets us perform experiments quickly and gain a solid empirical understanding of the different components of our method. Then, we apply it to three robotic environments where we have continuous high-dimensional observations and actions. In contrast to previous methods, our approach can deal with high-dimensional input directly without feature engineering. Finally, we apply it on the challenging Montezuma's Revenge Atari game which has visually rich pixel based observations and discrete actions. Despite their differences, our method manages to learn useful and interesting skills in all of them. This shows that neural reward functions are equipped to encode meaningful tasks in diverse environments. We highlight the main contributions of this paper:

- Propose the first skill-discovery algorithm to work in high-dimensional environments without prior expert knowledge.
- Solve a 2d maze that cannot be solved by using random exploration.
- Acquire complex skills including performing front- and back-flip in the HALF-CHEETAH, running in all directions in ANT and HUMANOID, standing and jumping on one leg in HUMANOID.
- Learn to run as fast in HUMANOID as a supervised agent trained for tens of millions of steps.
- Achieve a higher Particle-Based Mutual Information Metric (Gu et al., 2021) than approaches that explicitly optimize this metric.
- Collect the first key and reach several rooms in Montezuma's Revenge.

## 2 Related Work

Our work fits best into the unsupervised skill discovery literature (Mohamed & Rezende, 2015; Gregor et al., 2016; Florensa et al., 2017a; Achiam et al., 2018; Eysenbach et al., 2018; Sharma et al., 2019; Choi et al., 2021). Compared to DIAYN (Eysenbach et al., 2018) and similar approaches, we do not fix the number of skills to be learned at the beginning of training. This allows us to learn new skills in an open-ended fashion. On top of that, maximizing mutual information can lead to degenerate behaviors in high-dimensional environments. By manipulating a small subset of the dimensions, a lot of information can be encoded, without exploring the rest of the state space. Due to this issue, previous methods only consider a hand-picked subset of the dimensions to perform well.

Open-ended learning (Srivastava et al., 2012; Wang et al., 2019, 2020; Campero et al., 2020; Dennis et al., 2020; Ecoffet et al., 2021; Stooke et al., 2021) is closely related to unsupervised skill discovery. However, most approaches require either a parameterizable environment (Wang et al., 2019, 2020), some fixed encoding of tasks (Stooke et al., 2021) or self-competition (Silver et al., 2017a; Baker et al., 2019). This limits the applicability to environments that are engineered with these restrictions in mind. In contrast, neural networks are universal function approximators and thus, our approach can encode any possible task in any possible environment, as long as the input is chosen appropriately. Similar to our method, the Go-Explore algorithm (Ecoffet et al., 2021) also explores around the frontier of the known. It uses a handcrafted feature map to group similar states. Instead, we use neural reward functions to group them as skills. This does not use any expert knowledge.

Another related line of research to our approach is intrinsic motivation (Stadie et al., 2015; Bellemare et al., 2016; Pathak et al., 2017; Burda et al., 2018b,a; Pathak et al., 2019; Raileanu & Rocktäschel, 2020). These approaches have managed great success in hard-exploration Atari games. However, these approaches do not learn multiple skills by default. Additionally, their goal is to reward all novel states, which leads to general policies that visit many states. In contrast, skill discovery algorithms try to reward narrow regions of the state space to achieve controllability of meaningful dimensions.

Encoding reward functions as neural networks has also been considered in the literature. Compared to our work, they are trained with supervised signals (Abbeel & Ng, 2004; Fu et al., 2017; Singh et al., 2019; Li et al., 2021). Other approaches train auxiliary rewards with meta-learning (Zheng et al., 2018; Du et al., 2019; Veeriah et al., 2019) to enhance the learning of the original reward function.

Another approach to train multiple behaviors is goal-conditioned learning (Kaelbling, 1993; Schaul et al., 2015; Andrychowicz et al., 2017; Rauber et al., 2017; Nair et al., 2018; Veeriah et al., 2018;

Warde-Farley et al., 2018; Pong et al., 2019; Choi et al., 2021). In automated curriculum learning (Bengio et al., 2009; Florensa et al., 2017b; Forestier et al., 2017; Graves et al., 2017; Sukhbaatar et al., 2017; Florensa et al., 2018; Matiisen et al., 2019; Narvekar et al., 2020; Portelas et al., 2020a,b; Zhang et al., 2020), a sequence of goals is created such that each of them is not too hard nor too easy for the current agent. These approaches mostly rely on low-dimensional goal embeddings. When dealing with high-dimensional observations, they must use dimensionality reduction techniques. These techniques can introduce instabilities or destroy relevant information from the input. Our approach, on the other hand, can deal with high-dimensional inputs directly. On top of that, goals encode a narrow region of the state space, while each of our reward functions can be rewarding in a large region. This speeds-up the exploration in 'easy' regions of the state space.

## 3 Method

We introduce a method that performs open-ended, unsupervised skill discovery. It iteratively creates pairs of neural reward functions $R_\psi$ and policies $\pi_\theta$ trained to maximize the corresponding $R_\psi$. Our proposed method alternates between increasing the complexity of the reward function $R_\psi$ and leveraging the previously learned skills to learn a policy $\pi_\theta$ that can solve the new $R_\psi$. This yields a general learning procedure that learns complex skills in a diverse set of environments. See Figure 1 for a high level overview.

### 3.1 Increasing the complexity of $R_\psi$

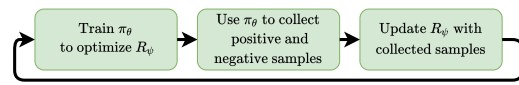

Figure 1: Main steps of our algorithms open-ended training loop.

The reward function in a fully observable Markov Decision Process (MDP) is by definition a function of the current observation, the next one and the action that was performed. However, in many cases this can be reduced to a function of just the current observation. Because of this, we opt for these simpler reward functions as the basis of our $R_\psi$. In our method, the reward function $R_\psi$ is a neural network which takes observations $o_t$ as input and outputs a single scalar value, the instantaneous reward $r_t^\psi$.

Assume that we have a policy $\pi_\theta$ that can reach states in the MDP which are rewarding under $R_\psi$. To increase the complexity of the reward function, we do the following:

- **Decrease the reward** of states which are visited by $\pi_\theta$, as we already have a policy that reaches these states. We create a data set $O_{neg}$ of such states. We refer to these states as *negative samples*.
- **Increase the reward** of states that can almost be reached by the current policy. This allows us to leverage $\pi_\theta$ to learn the new reward function. We create a data set $O_{pos}$ of such states. We refer to these states as *positive samples*.

To generate the negative samples, we run $\pi_\theta$ for a given number of steps (ideally until it reaches rewarding states) and store the visited states. To then generate positive samples, we change to performing random actions[2] for a fixed number of steps. To ensure that the new reward function is different from all previous ones, we also keep track of all the negative samples that we have collected for all skills in a data set $O_{neg\_all}$.

Finally, we set target values $a$ and $-a$ for the positive and negative samples respectively, and train the reward network using standard supervised learning on the following loss:

$$\mathcal{L}_\psi = \sum_{o \in O_{neg}} \frac{(R_\psi(o) + a)^2}{|O_{neg}|} + \sum_{o \in O_{neg\_all}} \frac{(R_\psi(o) + a)^2}{|O_{neg\_all}|} + \sum_{o \in O_{pos}} \frac{(R_\psi(o) - a)^2}{|O_{pos}|}$$

This loss ensures that positive samples that have never been seen before have positive reward in the next $R_\psi$, while all other samples that have been seen before decrease their reward. In the reinforcement learning phase we clip rewards to the $[0, a]$ range. This ensures that the agent seeks positive samples, rather than less negative ones.

---

[2]In MDPs with discrete actions we sample actions u.a.r. and in the continuous case, we keep the mean of $\pi_\theta$ and increase the standard deviation.

## 3.2 Forward transfer for $\pi_\theta$

Given the procedure presented in Section 3.1, we create increasingly complex reward functions. While this is great for open-ended learning, it eventually leads to skills that are too complex and cannot be learned from scratch. In order to learn these skills, we must leverage previous knowledge about the environment. In this section we present several forward transfer mechanisms that are necessary for the most complex skills.

In principle, our method can be combined with any standard reinforcement learning (RL) technique. But here, we focus on actor-critic methods like Advantage Actor-Critic (Mnih et al., 2016) or Proximal Policy Optimization (Schulman et al., 2017) for learning $\pi_\theta$ for several reasons. These methods have a value network that is separate from the policy which allows us to use different transfer mechanisms for the value and policy network. Also, these approaches work for both continuous and discrete action spaces. This allows us to use the same technique for the robotic environments and for the 2d navigation tasks. Finally the learned policies are stochastic which increases the diversity of negative samples and speeds up the skill discovery process. See Section 4.1.2.

We present our three forward transfer mechanisms below. All of them exploit the similarity between consecutive reward functions to ensure that even very complex reward functions can be solved by the RL agent in a reasonable number of environment interactions.

- **Value Reuse:** Initialize the value network to the final value network of the previous agent. While two consecutive reward functions are different, both still reward close-by regions of the state space. Thus, by keeping the previous value function, the policy network will be nudged towards that region of the state space from the very first gradient updates.

- **Policy Feature Reuse:** Initialize the policy network to the final policy of the previous agent but setting the weights of the final layer to 0. This keeps the previously learned features, but outputs a uniform policy over all actions (or mean 0 and a fixed standard deviation in the continuous case) which allows for proper learning and exploration[3].

- **Guiding Policy:** Act with the previous policy for a random number of steps at the beginning of each episode. This heavily simplifies the exploration problem. The agent will start exploring from states that are much closer to the rewards defined by $R_\psi$. This is because the previous policy could already solve the previous reward function. In contrast to the other two mechanisms, this one does not rely on initialization. Thus, this is the most effective in sparse reward functions that need many parameter updates to be learned.

In Section 4.1.1 we individually evaluate these three techniques and show that their combination is necessary in complex environments.

Putting everything together we get an algorithm which learns reward functions that encode increasingly complex behaviors and learns RL agents that solve those reward functions. Figure 1 illustrates the main steps of our training loop and see Algorithm 1 in the appendix for more detail.

## 4 Experiments

We now proceed to experimentally test our method. First, in Section 4.1, we thoroughly test all different components of our model in a 2d navigation task. This task allows us to verify the function of each component and also to explicitly visualize what each reward function is encoding.

Then, in Section 4.2, we move to BRAX robotic environments (Freeman et al., 2021). These have a lot of flexibility and thus allow the agent to learn very complex tasks. In these tasks, we evaluate the complexity of our skills by measuring their zero-shot transfer ability to the environment rewards. In the HUMANOID environment, our unsupervised skills outperform supervised agents trained for tens of millions of time steps. We also compute the one dimensional particle-based mutual information metric that has been proposed in the literature before (Gu et al., 2021) and show that our method outperforms previous approaches, even when other approaches only consider handcrafted feature dimensions in their objective.

---

[3]Policies at the end of learning can be very deterministic which slows down or completely stops learning of new reward functions.

Finally, in Section 4.3, we apply our method to Montezuma's Revenge. We show that the learned reward functions keep getting increasingly complex and we are mostly limited by the amount of compute that it takes to learn each new reward function.

## 4.1  2D Navigation Task

The task consists of a 32 by 32 maze with several walls and a 'danger zone'. The observation is given as a 32x32x1 image with all values set to 0, except a 1 in the current position of the agent. The agent starts in the top left corner and can perform 5 actions, either move in one of the cardinal directions or stay in the current position. If the agent moves into a wall it stays at its current position instead. If the agent is in the 'danger zone' and moves up, moves down or stays, the episode is terminated and the agent is moved back to the starting position. Figure 2 shows the layout of the maze. The 'danger zone' ensures that random exploration will not work to reach many parts of the environment and lets us easily test both the increasing complexity of the reward functions and the importance of forward transfer. We computed the expected number of steps to reach the bottom right corner with a random walk using Dynamic Programming. In expectation, $7 \cdot 10^{27}$ episodes are needed to do so. This shows that this maze is difficult to navigate.

To train our agent we use the Advantage Actor Critic (A2C) (Mnih et al., 2016) algorithm. To learn the rewards we use the full algorithm presented in Section 3. We use the same architecture for the reward, policy and value networks, but do not share any parameters. The architecture is a ReLU network with 2 convolutional layers followed by 2 fully connected layers. For the exact hyper-parameters see Table 3 in Appendix B. Figure 2 plots the most visited locations for each skill of one run of our algorithm. The first few skills visit points near the origin, later skills start moving to harder to reach parts of the state space. After roughly 40 iterations they reach the bottom right part. As stated before, this would take unreasonably long when using only random exploration.

Inspired by the BRAX library (Freeman et al., 2021), we implemented both the environment and an A2C agent inside a single JAX (Bradbury et al., 2018) compiled function. By doing this, the computation graph of the environment and agent are optimized jointly and both run on the GPU. This eliminates the need to send data between the CPU and GPU which is one of the main bottlenecks in RL. Using just one NVIDIA RTX 3090 GPU, the training process runs at over one million frames per second which enables training of agents in just a few seconds. This allows us to experiment quickly and at a lower economic and environmental cost. We believe this code is useful for the RL community on its own and provide it in the supplementary material.

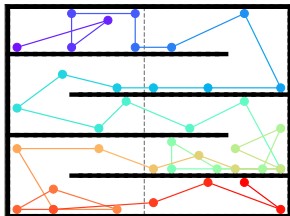

Figure 2: The black bars represent the walls and the dotted line represents the danger zone in the 2d maze. 40 skills discovered by our method are shown. For each skill, the circles represent the average position of locations visited often. Colors change from early skills in purple to late skills in red. Consecutive reward functions are connected by a line. The agent learns increasingly complex skills, until it reaches the bottom right corner of the maze; the hardest part to reach in the whole maze.

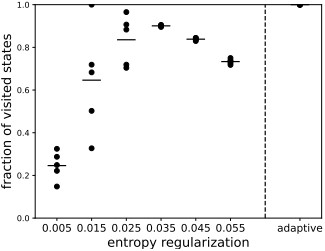

Figure 3: Total number of states visited by the learned policies after 60 generations as a function of the entropy regularization used by the underlying A2C agent. The adaptive entropy model manages to visit all states in all runs.

### 4.1.1  Forward transfer of skills

As pointed out in Section 3.2, our reward functions become too complex to be learned from scratch with random exploration in a reasonable number of steps. When this happens, our agent must rely on transferring knowledge from previous generations. Our navigation task is specifically designed to test this transfer ability, as random exploration would never reach the bottom right corner ($7 \cdot 10^{27}$ episodes in expectation).

We experimentally evaluate the three forward transfer mechanisms proposed in Section 3.2: **Value reuse**, **Policy feature reuse** and **Guiding policy**. The reward functions from Figure 2 serve as tasks,

sorted according to creation order. We train ablations of the three mechanisms sequentially on these tasks. This allows us to ignore the skill discovery process and only measure the forward transfer of skills. We repeat each experiment three times. The agent with all mechanisms always manages to solve[4] all reward functions. On the other hand, the Policy, Value and Guiding ablations fail to learn after solving $29.3 \pm 9.5$, $18.7 \pm 11.6$ and $19 \pm 8.5$ reward functions, respectively.

### 4.1.2 Speeding-up deep exploration

One key parameter when training actor critic methods is entropy regularization. In our method, policies with a lot of entropy generate a diverse set of negative samples. Diverse negative samples lead to reward functions that evolve more in each generation. This is especially beneficial in environments where many steps are necessary to reach certain states, like in this 2D navigation task or Montezuma's Revenge. We empirically verify this claim by training a set of agents with varying levels of entropy regularization. Figure 3 shows the coverage of the state space after 60 generations as a function of the entropy regularization. We observed that higher entropy leads to a faster coverage of the state space up to a certain threshold. However, too much entropy leads to policies that do not learn to reach the rewarding states when these are far away from the origin. Observe that this problem arises independent of the training procedure as entropy regularization changes which policy is considered optimal.

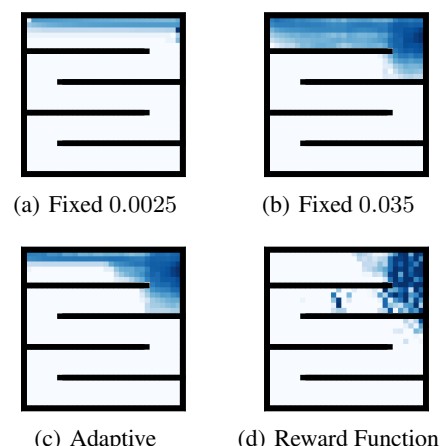

(a) Fixed 0.0025          (b) Fixed 0.035

(c) Adaptive          (d) Reward Function

Figure 4: States visited by policies trained to solve a reward function (d) that was created in the training process of Figure 2. We trained two policies with a fixed entropy regularization of $0.0025$ (a) and $0.035$ (b). (c) shows an agent trained with an entropy regularization of $0.0025$ in non-rewarding states and of $0.035$ in rewarding ones.

Hence, we want to have policies that gather diverse samples to change the reward function but can also be deterministic in parts of the trajectory that already have no reward. To achieve this, we use adaptive entropy regularization. That is, we apply a small entropy regularization term everywhere. We increase the entropy regularization in states where the reward function is positive, i.e. states in which we want to decrease the reward function. See Figure 4 for a visualization of the impact of the different entropy regularization strategies. With this, the method manages to consistently visit all states in the maze. This technique was necessary to create the reward functions from Figure 2.

### 4.2 Robotic Environments

Quantitatively measuring unsupervised reinforcement learning progress remains an open problem and active area of research (Gu et al., 2021). Because of this, it is still important to visualize and qualitatively study the learned skills. We do this in Section 4.2.1. In the next two Sections, 4.2.2 and 4.2.3, we quantitatively measure the performance of our algorithm on downstream task performance and with the so-called particle-based mutual information metric.

In contrast to the previous experiments, the action space is continuous and the input modality is now a vector of features, like relative position, angle and speed of the different joints. We do not use adaptive entropy here, as we need deterministic policies to maximize the downstream task performance and the mutual information metric. Given that the environments do not have far away regions to reach, we did not implement the guiding algorithm here. We do ablations for the other two transfer mechanisms. Note that the agent does not see the $x$-$y$ position. Details about hyper-parameters and architecture can be found in Appendix C.

In addition to skill discovery methods, we also compare our methods to RND (Burda et al., 2018b) and 'Disagreement' (Pathak et al., 2019), two of the most prominent works in intrinsic motivation. By default, these methods only create a single final policy. We extract multiple skills by taking

---

[4]We consider a reward function as solved if the agent manages to find positive reward.

checkpoints of the policy regularly during training [5]. To our knowledge, this has not been done before and we believe this showcases how closely related intrinsic motivation and unsupervised skill discovery are. We tuned the hyper-parameters of both methods extensively to maximize the Mutual Information metric (c.f. Section 4.2.3).

### 4.2.1 Qualitative Analysis

In Figure 5 we illustrate a selection of particularly interesting skills. Videos of these and additional skills can be found here. In Figure 6 we see how the velocity of learned skills in ANT and HUMANOID evolve over training. See Appendix E for the other runs. One can see that consecutive skills slightly change the direction and speed of moving. However, it is important to realize that the observations contain tens to hundreds of dimensions. Thus, the skills can encode much more complex behaviors than speed and direction of movement. For example, Figure 5 shows that the skill involves keeping one leg in the air on top of moving in the right direction.

We also show scatter plots for RND and 'Disagreement' in Appendix E. While the two methods also work well in the HUMANOID environment (c.f. Figures 14 and 15), they do not find any useful behavior in the ANT environment (c.f. Figures 11 and 12). These approaches reward all novel states, they lead to policies that cover large regions of the state space, as can be seen in ANT. We believe that the inherent instability of HUMANOID forces the policies to be more specific.

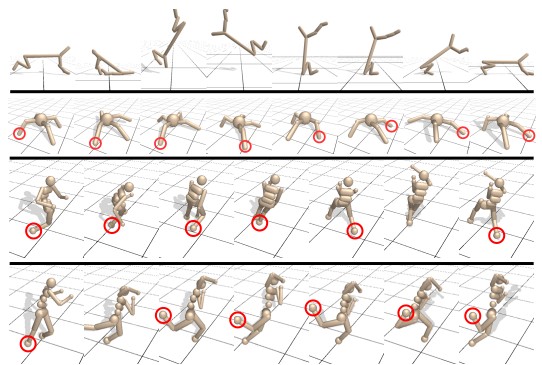

Figure 5: Visualization of four learned robotic skills. One of the legs is tracked with a red circle when it is visible. The HALF-CHEETAH learns to do a front-flip. The ANT does a partial rotation around its torso and then starts running. The upper HUMANOID also does a partial rotation and then runs backwards. The lower HUMANOID jumps on one leg while moving forward.

### 4.2.2 Zero-Shot Transfer

After discovering 50 unsupervised skills, we identify the skill that aligns best with the reward given by the environment. We report this zero-shot[6] performance in Table 1. As a baseline, we train an agent from scratch and measure how long it takes it to reach an equivalent performance. We use the Optuna hyper-parameter optimizer (Akiba et al., 2019) to find hyper-parameters which maximize the speed of learning beforehand. The exact procedure can be found in Appendix D.

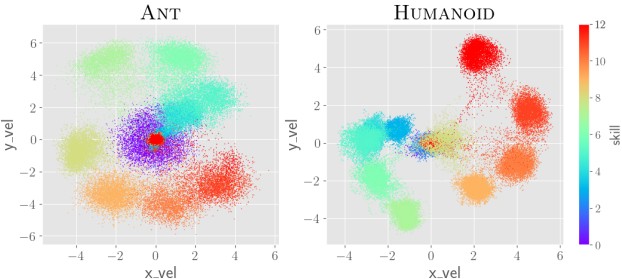

Figure 6: Scatter-plots of the $x$- and $y$-velocity of the states visited by the first 13 skills of one run in ANT and HUMANOID. The second half of the trajectory is shown. Colors change from early skills in purple to late skills in red.

The simplest environment, HALF-CHEETAH, benefits the least from unsupervised learning, while the hardest one, HUMANOID, benefits the most. We report the zero-shot performance of RND, 'Disagreement' and the ablations in Table 7 in the Appendix. As already seen in Section 4.2.1, the intrinsic motivation methods do not find any interesting behavior in ANT. Conversely, they perform well on the HUMANOID environment, the latter one even slightly outperforming our method[7].

---

[5] We thank anonymous reviewers 'Wvtf' and 'uUdP' for suggesting this baseline.

[6] Technically, we query the true reward function to identify the skills, but do not perform any training with it.

[7] Note that the tuning RND and 'Disagreement' for the $x$-velocity MI-metric promotes locomotion.

### 4.2.3 Particle-based Mutual Information metric

Measuring how well the agent can control relevant state dimensions is another way to track progress in Unsupervised reinforcement learning. This is measured using the mutual information between state dimensions and skills. While high-dimensional estimation of mutual information is an active area of research, sampling can be an effective form of estimation in the 1-dimensional case, c.f. Algorithm 1 in (Gu et al., 2021). We report the particle-based mutual information for the $x$-velocity in Table 2, using the same bucketing strategy[8] as in (Gu et al., 2021).

Table 1: Zero-shot environment reward of our algorithm and the number of steps a supervised PPO agents needs to match it. Both columns averaged over 10 repetitions.

| Task | Zero-shot reward | Steps from scratch |
|------|------|------|
| Cheetah | $1094 \pm 1130$ | $340K \pm 50K$ |
| Ant | $2506 \pm 511$ | $1.2M \pm 0.24M$ |
| Humanoid | $9092 \pm 1063$ | $55M \pm 27M$ |

Our method heavily outperforms DIAYN and DADS, when both methods look at the complete observation space. DIAYN and DADS achieve diversity by learning a set of skills that can be correctly labeled by a neural network. In high-dimensional spaces this is easy to do by relying on a small subset of all state dimensions. This leads to non-diverse behaviors across most state dimensions. Even when expert knowledge about relevant dimensions is supplied to other methods, i.e. only taking the $x$- and $y$-velocity into account, our method still fares well. Particularly, in the most complex environment, HUMANOID, our method performs best. We believe that the iterative increase in complexity leads to better coverage of hard to reach regions of the state like high speed or running backwards. With all this, our approach achieves greater controllability of the $x$-velocity than DIAYN and DADS without any kind of feature engineering.

Surprisingly, our novel[9] baselines, RND and 'Disagreement', also achieve better performance than DIAYN and DADS in the HU-MANOID environment, but not in ANT. As already discussed in Section 4.2.1, we believe that the larger instability of HUMANOID compared to ANT is the reason for this difference in performance.

Table 2: Particle-based mutual information metric for the $x$-velocity. Results are averaged over 10 runs. Algorithms with feature engineering only consider $x$-$y$ velocities. Baselines taken from (Gu et al., 2021)

| Task | Method | Feature Engineering | MI$(s, z)$ |
|------|------|------|------|
| Cheetah | Ours | ✗ | $1.40 \pm 0.21$ |
| Cheetah | DIAYN | ✗ | $0.49 \pm 0.16$ |
| Cheetah | **DIAYN$_p$** | ✓ | $\mathbf{1.82 \pm 0.20}$ |
| Cheetah | GCRL | ✓ | $1.63 \pm 0.16$ |
| Ant | **Ours** | ✗ | $\mathbf{1.33 \pm 0.11}$ |
| Ant | RND | ✗ | $0.26 \pm 0.15$ |
| Ant | Disagreement | ✗ | $0.08 \pm 0.03$ |
| Ant | DIAYN | ✗ | $0.07 \pm 0.01$ |
| Ant | DADS | ✗ | $0.32 \pm 0.06$ |
| Ant | DIAYN$_p$ | ✓ | $1.12 \pm 0.27$ |
| Ant | GCRL | ✓ | $1.22 \pm 0.19$ |
| Humanoid | **Ours** | ✗ | $\mathbf{1.29 \pm 0.25}$ |
| Humanoid | RND | ✗ | $0.94 \pm 0.17$ |
| Humanoid | Disagreement | ✗ | $1.05 \pm 0.14$ |
| Humanoid | DIAYN | ✗ | $0.07 \pm 0.01$ |
| Humanoid | DADS | ✗ | $0.24 \pm 0.06$ |
| Humanoid | DIAYN$_p$ | ✓ | $0.93 \pm 0.13$ |
| Humanoid | GCRL | ✓ | $0.77 \pm 0.15$ |

### 4.3 Montezuma's Revenge

To show the generality of our approach we evaluate it on the notoriously hard Montezuma's Revenge Atari game. In this game, the agent controls a character in a complex 2d world with several rooms. Appendix G shows the initial room and various items with which the agent may interact. Same as in (Mnih et al., 2015), the observation is a stack of the last 4 frames. This gives the agent information about speed and direction of movement. This is done for all networks, that is, the value, policy and neural re-

---

[8]We split the dimension in 1000 bins in the $[-10, 10]$ range. As in (Gu et al., 2021), we only take the second half of each trajectory. The initial state is independent of the skill and thus the beginning of the trajectory does not tell anything about the gained controllability.

[9]The approaches are not new, but using them to generate checkpoints that can be used as skills has not been done before to the best of our knowledge.

ward networks. We use the same simple CNN architecture as in (Mnih et al., 2015) for all three networks. For exact learning details see Appendix H.

Our algorithm uses finite episode lengths because once it reaches a rewarding state, the agent can stay there forever. Because of this, we reset the environment every $500$ steps.

One of the main difficulties when dealing with Montezuma's Revenge is that it cannot be simulated as fast as the other studied environments. On top of that, an agent can learn hundreds of *different* skills without ever leaving the first room. Finally, skills learned by our agent evolve from simple to very complex and extended in time. In the beginning, the agent just needs to stay close to the initial position. By the end of training, the agent learns to collect a key, open the door, avoid several enemies and visit four different rooms. In the most complex skills, it takes the agent several hundred steps to reach a rewarding state. This means that experiments can take several days to visit a different room.

In order to save computation, we adapt the number of training steps to ensure the agent has learned each skill. See Appendix F for details.

We found out that the first thing the agent learns is to get the life counter to $0$. Then it starts exploring the room, with $0$ lives already. This is because losing a life happens very easily during random exploration and because once the agent reaches $0$ lives, all future positive samples will have $0$ lives. This has two side effects. On the one hand, the initial $100 - 200$ steps of an episode are spent losing lives. On the other hand, exploration is harder, as the agent will reach a terminal state very easily. Because of this, we cut out the part of the image that shows the remaining lives, see Appendix G.

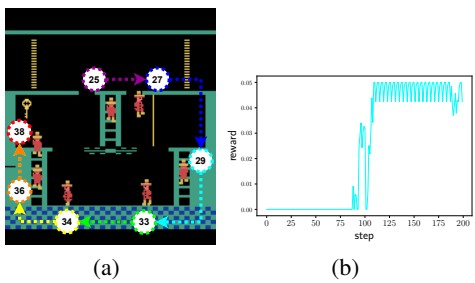

(a)                                    (b)

Figure 7: (a) The position of the agent in the most rewarding state of each skill. The skills are numbered by the creation order. (b) The rewards received by skill 29 during part of its trajectory. This shows the long term nature of the discovered skills.

Figure 7(a) visualizes different skills. The first observations which get the maximum possible reward are overlaid. All the shown skills go to the bottom and kill the skull first. Then, they move to a specific location. It can be seen that later skills explore states further and further away from the initial state. Figure 7(b) shows that the skill receives no reward for many steps. Only after the agent has killed the skull being on the ladder becomes rewarding[10]. This shows that both the reward network and the agent have learned a lot of the concepts that are necessary to tackle Montezuma's Revenge. This includes controlling the agent in the 2d environment, killing enemies and interacting with different objects. All of this without ever accessing the original reward function. Eventually, after learning a few hundred skills, the neural reward function pushes the rewarding states all the way to the fourth room. This video shows a run of this skill.

# 5   Limitations

Our approach has three main limitations: the randomized exploration strategy, the re-learning necessary for each new skill and the depth-first nature of the search of skills.

We have shown that the random exploration allows for discovering diverse skills. However, it can lead to slow progress in complex regions of the state space. Using more sophisticated exploration policies could lead to faster evolution of the skills. A potential avenue for improving this is using previously discovered skills as exploration policies.

As stated in Section 3.2, we use multiple techniques to make the learning of the next policy more efficient. Still, the vast majority of our compute is used for learning these policies. Using more advanced transfer learning or even meta-learning algorithms, could make the policy learning more efficient. Another possibility would be to relabel previously seen trajectories with ideas similar to (Rauber et al., 2017).

---

[10]Using the ladder is needed to kill the skull, but the reward function gives no reward when the agent is on the ladder before killing the skull.

Finally, our approach only uses the current skill to create the next one. On a high level, it uses a single search point at each time. Then, the iterative increase in complexity of the skills leads to a DFS-like exploration. This may not always be optimal. For example, in Montezuma's Revenge it causes the agent to explore only one of the two possible exits of the first room. On top of that, our algorithm can sometimes get stuck in places were backtracking is complicated or impossible, see Figure 9 in the Appendix. This also happens in Montezuma's Revenge when the life-counter is not removed. Figuring out smart back-tracking strategies or keeping multiple search points could address this problem and make our algorithm even more efficient.

## 6   Conclusion

We have presented an unsupervised reinforcement learning algorithm that uses reward functions encoded by neural networks. Our algorithm alternates between increasing the complexity of the reward function and transferring previous knowledge to learn a new skill that finds rewarding states. This allows it to learn an unbounded number of skills.

We have thoroughly tested the different components of our model in a 2d navigation task. This has allowed us to better understand our method in practice. We have shown that our method works both with high dimensional feature inputs, in robotic environments, and pixel inputs, in Montezuma's Revenge. Our algorithm learned a diverse set of skills in both settings. In HUMANOID and Montezuma's Revenge, skills found by our method achieve a zero-shot performance that takes millions of steps to learn in the classical reinforcement learning setup.

We believe our algorithm is one step in a direction that may one day allow reinforcement learning agents to fully understand an environment without making use of any predefined reward function. Just like in Computer Vision and Natural Language Processing, this will lead to agents that need very few labels from the task at hand to be able to solve it and will drastically expand the applicability of reinforcement learning.

## Acknowledgements

We would like to thank Xun Zuo for helping with the implementation of the maze environment, on top of many great discussions. We would also like to thank Frederik Benzing and Yassir Akram for many fruitful discussions. Finally, we also thank the anonymous reviewers for the thorough and insightful reviews and discussions. Robert Meier and Asier Mujika were supported by grant no. CRSII5_173721 of the Swiss National Science Foundation.

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
