# OpenReview forum: "Open-Ended Reinforcement Learning with Neural Reward Functions"
_NeurIPS.cc/2022/Conference — NeurIPS 2022 Accept_

### Official Review · Reviewer_uUdP · 2022-07-09

**Rating:** 5
**Confidence:** 3
**Soundness:** 3 good
**Presentation:** 3 good
**Contribution:** 3 good

**Summary:**

This paper presents a method for unsupervised skill discovery in reinforcement learning environments. The authors propose a method which iteratively learns policies for new skills by defining new reward functions for each new skill. New reward functions are constructed by taking the visited states by the previously learned skill, as well as all previously learned skills, and setting a low reward at those states. Rewards are higher for states which are close to being reached by the previous policy. The method takes advantage of some policy and value function transfer techniques to make learning more efficient. Experiments are conducted using a 2D navigation environment, robotics environments simulated using BRAX, and Montezuma’s Revenge.


**Questions:**

- Is the memory consumption of the method limiting? The reward function fitting seems to require keeping track of *all* of the previously seen negative states for all skills, which would grow as training continues. Additionally, as the number of negative states grows, does the objective in the supervised loss for the reward function need to be rebalanced for positive and negative classes?
- For Figure 2, I don’t quite understand what the meaning of the colored circles is. My understanding is that for each skill, a policy is learned which brings the agent from the top left further and further through the maze. But “The circles represent the average position of the most visited locations for each skill.” Does this mean that for a particular skill, the agent actually goes further into the maze than its circle is, and the circle just represents the average position for states visited during that trajectory?

Nitpicks/typos:
- 268: Unsupervised -> unsupervised
- Left quotes in several places in the manuscript are flipped
- Can the number of samples used to learn for Montezuma’s revenge/wall clock time be reported?


**Limitations:**

The limitations are mentioned in the conclusion, but it could be nice if the authors addressed the limitations of the current method more explicitly.

**Strengths And Weaknesses:**

Strengths of the paper:
- The method is straightforward and appears to make fewer assumptions than other methods like DIAYN do in terms of the x-y position prior, which is a very nice property.
- Evaluations are performed on sensible tasks and can obtain favorable results compared to DIAYN and GCRL, which is quite impressive given that the method is conceptually simple.
- Generally the paper is clearly written; I note some suggested small changes below, but there are some other minor grammatical errors which do not impact the clarity of the text but could benefit from another round of proofreading.

Weaknesses/clarifications:
- It seems that this work falls into the category of exploration with intrinsic curiosity. The reason is that while this method does generate pairs of discrete skills and policies whereas most exploration algorithms do not, the skills learned in this work aren’t necessarily disentangled. And while the authors note that “However, these approaches do not learn discrete skills that can be composed or fine-tuned for fast learning of new tasks”, it’s not clear from this work how the discrete skills learned here can be composed or fine-tuned for completing a new task. The zero-shot transfer task investigated in the experiments requires that a *single* learned skill can solve the entire task.
- As a follow-up to the previous point, it would be helpful to see more comparisons in the experimental section not only to additional unsupervised skill learning methods like DADS, but also intrinsic exploration methods like those mentioned in the related work section.

---

> ### Author Response · Authors · 2022-08-02
> **Response to Reviewer uUdP**
>
> We want to thank the reviewer for carefully going through our work and the useful comments. We try to address your concerns below point-by-point. We hope that these comments can lift some of the criticisms and lead the reviewer to lean towards accepting the paper. Of course, we are happy to respond to any additional questions during the discussion.
>
> **'It seems that this work falls into the category of exploration with intrinsic curiosity. '**:
>
> We agree that our method resembles work from intrinsic motivation. Though, we believe it is sufficiently different from the currently employed. To showcase this, we add a more detailed discussion of the differences to it and are implementing RND as a baseline. We refer the reviewer to the general response.
>
> **'...the skills learned in this work aren’t necessarily disentangled...'**:
>
> It is correct that our method does not directly optimize the disentanglement of the skills. The surprising thing is that our method produces skills which are more disentangled than skills learned by DIAYN. The mutual information metric reported measures the disentanglement of the skills and the x velocity. Our method outperforms the DIAYN baseline even when they use feature engineering. As mentioned in the general response, we are currently running the DADS baseline to strengthen this claim.
>
> **'...it would be helpful to see more comparisons in the experimental section not only to additional unsupervised skill learning methods like DADS...'**:
>
> We agree with the reviewer and will add the two already mentioned baselines.
>
> **'Is the memory consumption of the method limiting?'**
>
> This is a valid concern. In the robotic environments we only stored 1% of the total number of negative samples (cf.Table 6 in the Appendix). In Montezuma’s Revenge we only use the negative samples from the last 15 iterations (cf. Appendix H, line 661). As the agent always starts in the same position, negative samples are quite repetitive. Thus, it is enough to only store a fraction of all examples. We believe that one could reduce the number of negative samples stored even further.
>
> **'For Figure 2, I don’t quite understand what the meaning of the colored circles is.'**:
>
> We apologize that the explanation was not precise enough. For a specific skill we sample some number of trajectories. Then we only consider locations which were visited at least x times in total by this skill. After this, the remaining locations usually all have high reward.  Then, we take the average of these locations. This means yes, the agent goes a bit deeper into the maze than the circle. We will update the explanation to make it more clear.
>
> **'it could be nice if the authors addressed the limitations of the current method more explicitly.'**:
>
> We agree with the reviewer that limitations should be discussed more thoroughly. We will add a limitation section discussing and investigating the biggest limitation as outlined in the general response.

---

> > ### Comment · Reviewer_uUdP · 2022-08-09
> > **Thank you for the response**
> >
> > Thank you to the authors for the thorough response and clarifications to my questions. After reading the discussion, I'm inclined to agree with tSXg that checkpointing to extract skills from exploration style rewards is an interesting direction, to my knowledge unexplored in prior work. Since the authors have addressed many of my concerns and provided additional helpful baselines, I have increased my score.

---

### Official Review · Reviewer_Wvtf · 2022-07-09

**Rating:** 5
**Confidence:** 4
**Soundness:** 1 poor
**Presentation:** 2 fair
**Contribution:** 2 fair

**Summary:**

The paper studies the problem of unsupervised reinforcement learning, where an agent aims to explore its environment without a reward signal and learn diverse skills. The proposed approach amounts to learning a reward function which encourages exploration, and a policy with A2C to maximize this reward. The exploration reward function itself is essentially a classifier trained to predict visited states (negatives) from states near the boundary of the visited states (positives). The policy is trained to maximize the score (go near the boundary).

To recover "skills", the policy checkpoints at various stages of training are used. At each phase, the reward function then policy are trained using the newly collected data. To stabilize learning, the policy/value function are reused from the previous skill in learning the new skill.

Experiments show this can explore a maze environment as well as ant/humanoid environments. The proposed approach outperforms DIAYN in terms of mutual information between skill and state-distribution.

**Questions:**

- If the policy at different stages of training are taken as "skills", why are all of the listed intrinsic motivation papers not valid comparisons?
- How does the proposed method for state coverage differ/compare to the relevant work? There should be comparisons to at least some recent work which also does exploration+skill discovery by optimizing for state coverage.

**Limitations:**

Limitations are not really discussed.

**Strengths And Weaknesses:**

*Strengths*
- The paper studies an important problem in unsupervised RL and skill discovery.
- Expanding the policy to cover the state space is a well motivated objective for unsupervised RL.

*Weaknesses*
- First, the way the proposed method decomposes "skills" is a bit unclear. From what I can tell, the method is training a reward to explore the boundary of states, and a policy to maximize that reward, both being updated online. Then "skills" as denoted in this work are just the policy/reward at different checkpoints in training. So it is less that the method learns a multi-skill conditioned policy, but rather it is a collection of checkpoints (which get manually selected during downstream 0-shot adaptation).

If this is the case, then the claim in the related work that intrinsic motivation papers are not comparable is not quite accurate.
"Another related line of research to our approach is intrinsic motivation (Stadie et al., 2015; Bellemare et al., 2016; Pathak et al., 2017; Burda et al., 2018b,a; Raileanu & Rocktaschel, 2020). These approaches have managed great success in hard-exploration Atari games. However, these approaches do not learn discrete skills that can be composed or fine-tuned for fast learning of new tasks."
Any of these works could also produce skills via the same checkpointing procedure used here and thus are valid comparisons.

- Moreover, the core exploration objective proposed in this work is not particularly new compared to prior work. Theres a large literature of works that use state-space coverage as their exploration+skill learning objective, including model error/disagreement [1,2,3], pseudo counts [5,6], state distribution matching [7], and exploring the boundary of states [4,8] to name a few. While certainly not all are necessary, the paper should at least compare to some of these baselines, and currently none are. And the paper should do a better job of explaining how exactly their instantiation is different from what is being proposed in these works.

1. Pathak et al. Curiosity driven Exploration by Self-Supervised Prediction.
2. Pathak et al. Self-Supervised Exploration via Disagreement.
3. Sekar et al. Plan to explore vis Self-Supervised World Models
4. Mendonca et al. Discovering and Achieving Goals via World Models
5. Tang et al. # exploration: A study of count-based exploration for deep reinforcement learning.
6. Burda et al. Exploration by Random Network Distillation.
7. Lee et al. Efficient exploration by state marginal matching.
8. Bharadhwaj et al. LEAF: Latent Exploration Along the Frontier.

- Finally, there are a number of design choices in the proposed approach that are unclear. For example, why does random exploration from the current policys states produce a good set of positive examples for exploring beyond it. I can think of cases (like going through a narrow gap), where such exploration will not hit the states relevant for pushing the boundary. Approaches like those used in [4,7,8] seem like they could be more effective are identifying the boundary to explore.

---

> ### Author Response · Authors · 2022-08-02
> **Response to Reviewer Wvtf**
>
> We want to thank the reviewer for carefully going through our work and the feedback. In the following, we try to further clarify some aspects of our method. We hope that we are able to lift some of the criticisms and highlight some positive aspects of our work. Of course, we are happy to respond to any additional questions the reviewer has.
>
> **'From what I can tell, the method is training a reward to explore the boundary of states, and a policy to maximize that reward, both being updated online.'**:
>
> The reviewer is right to point out that our method has quite some resemblance to methods used in intrinsic motivation like Random Network Distillation. What makes our method differ from them is that we do not update our reward network online. We apologize that we did not make this clear enough in the paper and will update it accordingly. Our method takes turns in learning a new reward function until convergence. It uses a previously sampled data set of experience. Then, we train the next skill for many gradient steps on the fixed reward function.
>
> As already discussed in the general response, we believe that this approach has some advantages for learning skills. In particular, our reward function correspond to a specific region in state space, cf. Figure 4d). When using intrinsic motivation, everything which the agent has not seen would still be rewarding. This means that check-pointing methods from intrinsic motivation would not yield reward functions which only correspond to the current learned skill. We will update the discussion of our method in the paper to distinguish ourselves more from these works.
>
> **'While certainly not all are necessary, the paper should at least compare to some of these baselines, and currently none are.'**:
>
> As already discussed above, we will highlight the difference from our method to these intrinsic motivation work more clearly. We are working on adding a RND baseline for the robotic environments. To the best of our knowledge, none of these intrinsic curiosity approaches have been shown to work on a robotic task with a similar number of joints as the Humanoid environment. If the reviewer is aware of such work, we would appreciate a pointer to it.
>
> **'So it is less that the method learns a multi-skill conditioned policy'**:
>
> We believe that it is also a valid choice in skill discovery to have independent policy networks for different skills. It is unclear that having a single skill conditioned policy is actually advantageous for previous methods like DIAYN or DADS. In the paper Wasserstein Unsupervised Reinforcement Learning (cf. https://ojs.aaai.org/index.php/AAAI/article/view/20645 ) they note that both DIAYN and DADS performed better when the actor networks are independent. Additionally we want to point to Enhanced POET (cf. http://proceedings.mlr.press/v119/wang20l/wang20l.pdf ) where they also use a collection of independent actors to learn very complex behavior.
>
> **'For example, why does random exploration from the current policys states produce a good set of positive examples for exploring beyond it.'**:
>
> The reviewer raises a valid concern whether random exploration from the current boundary is enough. We believe that it is an interesting line of investigation. As our method has completely decoupled reward training and policy training, the exploration for the reward training can be substituted by different methods. In this work we chose the simplest exploration strategy possible. Even with random exploration, our method is able to find skills which need a precise sequence of action to not fail. This can be seen in all environments we considered including the death zone in the 2d grid. Random exploration is able to push the boundary a few actions at a time which then can be learned by our policy. We also added a 2d grid experiment with a narrow gap to showcase that our method can learn to go through these, cf Figure 11c) in Appendix I.
>
> As already stated above, we are happy to discuss any additional questions the reviewer has about specific design choices of our method.

---

> ### Comment · Reviewer_Wvtf · 2022-08-08
> **Re Response**
>
> Thanks to the authors for the additional information.
>
> > The reviewer is right to point out that our method has quite some resemblance to methods used in intrinsic motivation like Random Network Distillation.... Our method takes turns in learning a new reward function until convergence. It uses a previously sampled data set of experience....
>
> Got it, I understand that in the proposed approach that you are iteratively learning a reward function from scratch with negatives as seen data and positives as random exploration from the boundary. And I agree that implementing it this way with re-learning a new reward and policy from scratch iteratively is different than the way it is implemented in prior work. My point is more that at convergence this objective is maximizing state-space coverage - something that many prior works also optimize for. And many of those works [4, 7] both maximize state space coverage *and* learn skills in some form, either through a mixture of policies or through goal-conditioning.
>
> > As already discussed in the general response, we believe that this approach has some advantages for learning skills. In particular, our reward function correspond to a specific region in state space, cf. Figure 4d). When using intrinsic motivation, everything which the agent has not seen would still be rewarding. This means that check-pointing methods from intrinsic motivation would not yield reward functions which only correspond to the current learned skill.
>
> I'm not fully convinced by this line of reasoning. Again, the way the proposed approach gets "skills" is by saving model checkpoints. I agree this is a totally valid way of getting skills, but it is also applicable to *all* prior methods as well. And intuitively I would expect that saving checkpoints in this way for any of the exploration methods I listed in the original review would probably also give skills which correspond to different regions of the state space. For methods like [7] this is almost certainly the case since they use a mixture of policies. And again I think this is something that would be addressed by more comprehensive experiments.
>
> > As already discussed above, we will highlight the difference from our method to these intrinsic motivation work more clearly. We are working on adding a RND baseline for the robotic environments...
>
> I think it's great the authors are adding an RND baseline. I would also add that there are much more recent and better performing approaches like SMM [7] and LEXA [4], that have been shown to work on challenging robotic control tasks from pixel inputs, and also on real robots. But I understand that adding these is not feasible in the timescale of the rebuttal.
>
> > We believe that it is also a valid choice in skill discovery to have independent policy networks for different skills. It is unclear that having a single skill conditioned policy is actually advantageous for previous methods like DIAYN or DADS...
>
> I think this makes sense, and I think it's fine for different checkpoints to be "skills". My concern was more that if this is how skills are recovered, this same process can be applied to pretty much any prior work including the intrinsic motivation papers. Thus they are valid baselines, and the claim in the original work that "However, these approaches do not learn discrete skills that can be composed or fine-tuned for fast learning of new tasks." is not accurate.
>
> > The reviewer raises a valid concern whether random exploration from the current boundary is enough. We believe that it is an interesting line of investigation. As our method has completely decoupled reward training and policy training, the exploration for the reward training can be substituted by different methods....We also added a 2d grid experiment with a narrow gap to showcase that our method can learn to go through these, cf Figure 11c) in Appendix I.
>
> Agreed this is an interesting direction for investigation. My sense is that this is perhaps where model error might be a better indicator of the frontier of exploration, as is done in works like LEXA [4] or ICM [1].
>
> Overall, I appreciate the authors responses in the rebuttal. However I still find the positioning of the paper with respect to the prior work, and the missing comparisons to newer works a major limitation, and lean towards rejection. I do think the ongoing RND experiment is a great first step towards addressing this. I think if the paper can better explain (and show empirically) why its exploration objective is better than similar methods like SMM [7], or Plan2Explore/LEXA [3,4], I think it'd make for a good paper.

---

> > ### Author Response · Authors · 2022-08-09
> > **Additional response to Reviewer Wvtf**
> >
> > We thank the reviewer for their additional response. We are happy to hear that our responses and additional experiments are improving the paper. In light of the new results we want to respond to a few of the points you raised.
> >
> > > And intuitively I would expect that saving checkpoints in this way for any of the exploration methods I listed in the original review would probably also give skills which correspond to different regions of the state space.
> >
> > By design, these exploration methods cover a wide region of the state space. This is because they reward all states that were not visited yet and thus many different parts of the state space are rewarding. On the contrary,our approach only rewards a specific and different region of the state space at each iteration. This can also be seen in the new plots for the RND baselines in Appendix J. Especially in the Ant environment each checkpoint covers a very large part of the state space and different checkpoints overlap to a very high degree.
> >
> > > I think it's great the authors are adding an RND baseline. I would also add that there are much more recent and better performing approaches like SMM [7] and LEXA [4] …
> >
> > We decided to implement RND because RND is probably the most prominent recent intrinsic curiosity method and has been reused by other researchers to yield impressive results in large scale experiments, e.g. Agent57 and NeverGiveUp.
> >
> > > My concern was more that if this is how skills are recovered, this same process can be applied to pretty much any prior work including the intrinsic motivation papers.
> >
> > > and the claim in the original work that "However, these approaches do not learn discrete skills that can be composed or fine-tuned for fast learning of new tasks." is not accurate.
> >
> > No previous work that we are aware of used this checkpointing mechanism for skill discovery. Thus, the checkpointing is a modification of previous work. This checkpointing method could also be used as a baseline for all other skill discovery works. We propose a method that connects objectives similar to intrinsic motivation and skill discovery. But, we agree with the reviewer that the comparison to checkpointing these prior work is interesting. This is why we added the RND checkpoint baseline and believe that the results highlight the differences to mere checkpointing.
> >
> > To our knowledge, our work is the first one to showcase good performance on the Humanoid environment without any form of feature engineering. The checkpointing algorithm also outperforms DIAYN and DADS. This is also an interesting finding on its own. But, as already discussed, our method has additional advantages over the mere checkpointing. We hope that our response and the new results can improve your evaluation of our work.

---

> > > ### Comment · Reviewer_Wvtf · 2022-08-09
> > > **Re additional response**
> > >
> > > > By design, these exploration methods cover a wide region of the state space. This is because they reward all states that were not visited yet and thus many different parts of the state space are rewarding. On the contrary, our approach only rewards a specific and different region of the state space at each iteration. This can also be seen in the new plots for the RND baselines in Appendix J. Especially in the Ant environment each checkpoint covers a very large part of the state space and different checkpoints overlap to a very high degree.
> > >
> > > I looked at the new plots in the appendix, and for Ant it does seem the skills are better clustered than for RND.
> > >
> > > > No previous work that we are aware of used this checkpointing mechanism for skill discovery. Thus, the checkpointing is a modification of previous work. This checkpointing method could also be used as a baseline for all other skill discovery works. We propose a method that connects objectives similar to intrinsic motivation and skill discovery. But, we agree with the reviewer that the comparison to checkpointing these prior work is interesting. This is why we added the RND checkpoint baseline and believe that the results highlight the differences to mere checkpointing.
> > >
> > > Ok this makes sense.
> > >
> > > Overall the above responses do address my main concerns about the contributions of the work, and I'll raise my score to a borderline accept. I would strongly encourage the authors to add a comparisons to the SOTA approaches I listed earlier in the camera ready if the paper is accepted.

---

### Official Review · Reviewer_tSXg · 2022-07-09

**Rating:** 6
**Confidence:** 4
**Soundness:** 3 good
**Presentation:** 3 good
**Contribution:** 3 good

**Summary:**

The authors propose a method for unsupervised skill discovery via reinforcement learning. The method involves training a sequence of neural network-based reward functions, and using RL to learn policies that optimize these reward functions.  The reward functions are trained using positive and negative examples, where negative examples are the states reached by current policy, and positive examples are states reached by executing random actions. This encourages learning skills that cannot be visited by the current policy (and all previous policies). The authors apply this method for learning dozens of skills in simulated physics environments (half-cheetah, humanoid etc.), and hundreds of skills in Montezuma’s revenge. The skills often (but not always) recover interesting behaviors, including a humanoid hopping on one leg and the agent in Montezuma’s revenge being able to explore new rooms (beyond the first room).

**Questions:**

Questions
- “In expectation, 7 · 10^27 episodes are needed to do so. This shows that this maze is difficult to navigate.” How did the authors arrive at - this number?
- The ablations were only performed for the 2D navigation domain. Is there a reason why they were not done for the robotics domain? Since the two domains are fairly different, it is not necessary that all conclusions from the 2D navigation experiments will hold.

**Limitations:**

The paper lacks section on limitations (there is maybe only one or two lines in the conclusion that mention limitations, and they only do so in the context of future work that could be done). I would encourage the author to come up with a more comprehensive discussion of limitations (I mention some in the weaknesses above, but I would guess the authors are probably aware of more limitations).

**Strengths And Weaknesses:**

Strengths
- Simple, straightforward method.
- Impressive results in a few different domains.
- Clear writing and presentation.

Weaknesses
- The authors only show results on zero-shot transfer (i.e. how well a learned skill performs on some standard reward function), and there are no experiments on using the learned skill(s) for quickly learning a new task (i.e. improving beyond the pre-trained skill).
- The method is prone to learning uninteresting skills / getting stuck in local minimas. For example, in the experiments for Montezuma’s revenge, the authors noted that one of the first skills learned by the agent was to get the life counter to zero, which made it very difficult to do any further exploration (since the episode can terminate very easily once the life counter is at 0). This led the authors to manually intervene in the learning process and hide the part of the screen that shows remaining lives.

---

> ### Author Response · Authors · 2022-08-02
> **Response to Reviewer tSXg**
>
> We want to thank the reviewer for carefully going through our work and the overall good feedback. We will try to address your remaining concerns below point-by-point. We hope that these comments can lift some of the criticisms. Of course, we are happy to respond to any additional questions during the discussion
>
> **'The authors only show results on zero-shot transfer...'**:
>
> We agree that tuning/composing our learned skills for downstream tasks is another important use case. In the considered task the zero shot performance is already strong, hence we did not fine-tune it. Because we do not use any feature engineering, our method is not inherently biased to perform well on this specific task.
>
> **'The method is prone to learning uninteresting skills / getting stuck in local minimas...''**:
>
> We agree that this is the major drawback for our current method. As outlined in the general response, we will add a new section discussing and investigating this issue. In particular, we already added additional experiment plots into Appendix I to showcase this issue. These pinpoint the exact scenario where our method fails, i.e. when there is a region of states from which backtracking is impossible.
>
> **'“In expectation, 7 · 10^27 episodes are needed to do so. This shows that this maze is difficult to navigate.” How did the authors arrive at this number?'**:
>
> To come to this number we used dynamic programming. We computed the random walk state distribution after n+1 steps with the distribution after n steps. With this we can compute the probability that the agent reaches the bottom right until the end of one episode.
>
> **'Is there a reason why they were not done for the robotics domain?'**:
>
> We originally only ran the ablations on the 2D navigation domain because due to computational constraints.  We agree with you that the robotic domain is quite different, hence we also ran the value and policy ablation for Ant and Humanoid. We point to the general response for the results.
>
> **'The paper lacks section on limitations'**:
>
> We agree that it makes sense to discuss limitations more thoroughly. We believe that the failure mode you mentioned above is the major limitation. We point to the general response for an overview of how we investigate this more closely.

---

> ### Comment · Reviewer_tSXg · 2022-08-04
> **Thank you for the response.**
>
> Thank you for the response. I think the planned edits will improve the paper.
>
> I also took a look at some of the more negative reviews, and it appears that the main point of contention is that the work is very similar to prior work in exploration objectives for RL (such as RND, ICE etc.). I agree that this is the case, but, to the the best of my knowledge, this is the first work that extracts several skills from an environment using exploration-style objectives for reward learning, which makes this paper interesting. In a way, this paper ties some of the research threads in unsupervised skill discovery and intrinsic motivation / exploration, and while this connection seems a little obvious in retrospect, I haven't seen this connection explored before. As a result, I plan to keep my rating.

---

### Official Review · Reviewer_GsFo · 2022-07-11

**Rating:** 6
**Confidence:** 4
**Soundness:** 4 excellent
**Presentation:** 4 excellent
**Contribution:** 3 good

**Summary:**

The paper introduces a method for unsupervised skill discovery using learned reward functions. Pairs of reward function and policy pairs are iteratively learned. The reward function evolves by predicting negative rewards for states the agent has visited and positive rewards for states just outside the agent's reach. The policy is then transferred to the new reward using three forward skill transfer methods.

This new method is the first unsupervised skill discovery approach to work in high-dimensional environments without expert knowledge. The paper first analyzes a 2d maze task. Next, the paper shows results in continuous control tasks and compares to 3 baselines. Finally, the paper shows results on Montezuma's revenge from pixel inputs.

**Questions:**

* Why use a regression loss for training the reward function? Isn't this a classification problem for distinguishing the already visited negative states from the new positive states?
* I do not understand the justification for clipping the rewards to [0,a]. Could the negative rewards be important penalties for avoiding exploring parts of the state space that were already visited?
* How does the training compare to the baselines from Gu et al. 2021? Do these baselines also do such extensive hyperparameter tuning to make this comparison fair?
* What is the strategy used to adapt the number of training steps to ensure each skill has been learned in Montezuma's revenge?

**Limitations:**

The authors mention limitations of the work with the potential for better forward transfer methods and using a population of skills rather than the narrow deep search strategy.


**Strengths And Weaknesses:**

Strengths:
* To the best of my knowledge, learning a reward function for unsupervised skill discovery is novel. I believe this is an important step forward from prior work that relies on more hand-crafted reward objectives.
* Unlike other skills discovery methods like DIAYN, the proposed method does not require engineering only part of the observation space as input for good performance.
* Method demonstrates superior performance over relevant unsupervised skill discovery algorithms both in terms of the MI metric.
* Qualitative analysis demonstrates the behavior of skills learned in all three benchmarks.
* The algorithm does not need a specified number of skills ahead of time. New skills can continuously be added.

Weaknesses:
* While this weakness was also acknowledged in the limitations section, the proposed method learns skills in a depth first rather than breadth first way. This narrow search of skills could result in lacking skill diversity. Imagine a version of the 2d maze environment where there are multiple possible paths. I think the proposed method would find skills that only go along one path, but never discover skills down the other path because the skills progressively build off each other.
* Several design decisions are not quantitatively evaluated. The impact of adaptive entropy regularization and the three forward transfer methods are not ablated. Furthermore, there are several arbitrary hyperparameters related to the guiding phase like the duration of the guiding phase and the time spent collecting negative actions.
* Only the continuous control results include a comparison to baselines. It is unclear if the method helps in other tasks like maze navigation or Montezuma's revenge.

---

> ### Author Response · Authors · 2022-08-02
> **Response to Reviewer GsFo**
>
> We want to thank the reviewer for carefully going through our work and the overall good feedback. We will try to address your remaining concerns below point-by-point. We hope that these comments can lift some of the criticisms. Of course, we are happy to respond to any additional questions during the discussion
>
>  **'the proposed method learns skills in a depth':**
>
> This is a valid concern about our method. As outlined in the general response, we will add a more detailed discussion of this limitation. We especially refer to the new plots we added to Appendix I which investigate this limitation.
>
> **'The impact of adaptive entropy regularization and the three forward transfer methods are not ablated.'**:
>
> We ablate this mechanism in the 2d maze where we can run enough seeds to properly evaluate the choices made. Additionally we ran the policy and value transfer ablations for the ant and humanoid environments, cf the Table in the general response.
>
> **'Furthermore, there are several arbitrary hyperparameters related to the guiding phase like the duration of the guiding phase and the time spent collecting negative actions.'**:
>
> Yes, these choices are not properly evaluated. Due to computational constraints, we can not tune these parameters. E.g. in the robotic environments we fixed the duration of both phases at the beginning and only evaluated the choice which is in the final experiments. We believe that at least in the robotic environment the exact choices do not matter too much. Essentially all skills exhibit very repetitive behaviors and thus there are many very similar states in the trajectories.
>
> **'It is unclear if the method helps in other tasks like maze navigation or Montezuma's revenge.”**:
>
> We want to point out that previous skill discovery literature like DIAYN or DADS already relied heavily on feature engineering in the robotic environments. Most commonly decrease the dimension to two. Pixel based environments have very high-dimensional states. Thus, these methods are impractical for environments like Atari.
>
> **'Why use a regression loss for training the reward function?'**:
>
> You raise a valid point. We never tried it with a classification task and used regression from the beginning. We believe that both should work.
>
> **'Could the negative rewards be important penalties...'**:
>
> We agree that it would be great to be able to penalize old states with negative rewards. Unfortunately, negative rewards generally lead to degenerate behavior if the agent can terminate the episode itself by dying. It learns to die as quickly as possible. This is why many environments, as e.g. the original Humanoid give a small positive reward at each timestep.
>
> **'Do these baselines also do such extensive hyperparameter tuning...'**:
>
> Our hyperparameter tuning for the base PPO learning algorithm is of similar scale than the one used in Gu et al. 2021. Unfortunately they do not report how extensively they tuned the method specific hyperparameters.
>
> **'What is the strategy used to adapt the number of training steps to ensure each skill has been learned in Montezuma's revenge?'**:
>
> We refer the reviewer to Appendix F where we explain the strategy in detail.

---

> > ### Comment · Reviewer_GsFo · 2022-08-06
> > **Response to Authors**
> >
> > Thank you for the responses and added material. Reviewing the other reviews, I see other reviewers had concerns about connections to intrinsic motivation (IM) approaches. I agree that the paper needs to improve the discussion on the connections to IM approaches since that line of work appears similar. However, I agree with the authors and tSXg that this paper presents an interesting connection between IM and unsupervised skill discovery. To my knowledge, no prior work has demonstrated how IM-like objectives can extract skills for downstream tasks. IM objectives have only been shown with learning a single policy.
> >
> > How are the authors comparing to the RND baseline? The paper states "However, these approaches do not learn discrete skills that can be composed or fine-tuned for fast learning of new tasks." Has RND been modified to save checkpoints as skills like the proposed method?

---

> > > ### Author Response · Authors · 2022-08-08
> > > **Additional Response to Reviewer GsFo**
> > >
> > > We thank the reviewer for their additional response. We will update the paper’s related work section to highlight the similarities and distinction to work in intrinsic motivation.
> > >
> > > Yes, we modified RND such that we save checkpoints every 20M environment steps which is the same number of steps we train each of the skills in our method. We ran a hyperparameter search for the RND specific hyperparameters and are now evaluating the best found hyperparameters with more seeds. Our preliminary impression is that this works to some extent, i.e. this method extracts different behaviors at different points in time. But, it seems that the policies are less narrow than the ones learned by our method. It looks like the MI metric is considerably lower than our method but higher than the DIAYN basline. We will report our detailed results and add plots like Figure 6 by tomorrow.

---

### Author Response · Authors · 2022-08-02
**General response to all reviewers**

General response:
We thank all the reviewers for their detailed and constructive comments on the paper. We worked intensively (and still are) on addressing the reviewers suggestions and criticism which we address in the individual responses. Here we discuss new additional experiments and respond to some points raised by multiple reviewers. We outline what we will change in the manuscript. We hope that these additional experiments and changes to the manuscript will lead the reviewers to vote for accepting our work. We are happy to engage in additional discussions with the reviewers. Here is a short overview of what we add before we discuss each point separately:
- We are adding DADS as an additional baseline for the robotic environments.
- We perform ablations for the forward transfer mechanisms in the robotic environments
- We are implementing Random Network Distillation for the robotic environments and will report our findings as soon as we have them.
- We ran additional experiments on modified versions of the 2d maze to investigate how narrow our skill search is.
- We will add a more extensive discussion of limitations to the paper.


**DADS  baseline**

We are currently running DADS experiments on both the Ant and the Humanoid environment. We will post the results as soon as possible during the discussion phase.

**Ablations for the robotic environments**


| Method | MI-metric | zeroshot transfer |
|--------|----------|----------|
| Ant full method  | 1.33 ± 0.11    | 2506 ± 511     |
| Ant policy ablation | 1.11 ± 0.15    | 1716 ± 628     |
| Ant value ablation  | 1.23 ± 0.18    | 1924 ± 651     |
| Humanoid full method    |1.29 ± 0.25    | 9092 ± 1063     |
| Humanoid policy ablation    |1.01 ± 0.08 | 8913 ± 236|
| Humanoid value ablation    |0.91 ± 0.24 | 7361 ± 959 |

In the robotic environments we ran the ablation for the two forward transfer mechanisms on Ant and Humanoid. We report the results for five seeds here and are running five additional seeds to match the ten seeds used in the original experiment. The forward transfer mechanisms seem to also help in the robotic environments. The MI-metric of all ablations is lower, but the magnitude of the impact depends on the environment.

**New section ‘Limitations’**

The largest limitation of our current method is the narrow deep search we are conducting. If the method comes to a state where there are two possible paths, it explores along one of the paths. The drawback is that it potentially never returns to explore the other path. To investigate this, we ran additional experiments in different 2d grids. We added the figures to Appendix I such that the reviewers can view them.

First, we investigate what happens when the two paths are dead ends. In each iteration, the reward training increases the reward in some region. As can be seen in Figure 11a), this leads the agent to backtrack from a dead end. While backtracking is not very efficient, our method is able to deal with dead ends. We also saw similar backtracking in our experiments in Montezuma’s Revenge.

A more challenging scenario is when backtracking is not possible. A good example for this is losing a life in Montezuma’s Revenge. The agent can not gain lives back. To simulate this, we added traps in both possible paths, i.e. as soon as the agent crosses a line, it can not cross the line again to go back. See Figure 11b) for the result. The second time a skill goes into one of the traps the method does not recover and all future skills wander into the trap.

These two experiments characterize the failure mode of our method more precisely. We believe that using a population of skills to search from, can help to combat this issue as it is possible to explore both paths simultaneously.

**Random Network Distillation and other intrinsic curiosity methods**

Multiple reviewers pointed out the relation to intrinsic motivation and asked about comparisons to these works. We are implementing Random network distillation (RND) for the robotic environments and will report the result as soon as possible.

The major difference from our method to RND and other intrinsic motivation is that we do not train the reward module online. Our method takes turns of learning a reward function and a policy. We will update the related work section to make this important difference more clear.

Compared to the intrinsic motivation method, each of our reward functions reward a specific region of the state space instead of everything the agent has not seen yet. This can be seen in Figure 4d). Methods like RND would also reward all states deeper in the maze. Our more narrow reward functions encode more specific behaviors/skills. Due to this, we can train on a specific reward function for many steps to master some specific behavior.

Finally, to the best of our knowledge, these methods have not been showcased to work on robotic environments of similar scale as the Humanoid environment.

---

> ### Author Response · Authors · 2022-08-09
> **Second general response to all reviewers**
>
> Since our first response we worked hard on preparing additional results for the paper. Here we post these results here and discuss them. We will add these new results to the paper. Additionally, we will adapt the text of the paper according to the useful feedback of the reviewers. This includes adding the Limitations section and updating the Related work section to discuss the similarities and differences to work in Intrinsic motivation in more detail. We will also take into account the smaller suggestions made by individual reviewers.
> Due to time constraints, we cannot update the manuscript until the end of the rebuttal period but will work on it in the coming weeks. Nonetheless we hope that the strengthening of our results by the additional experiments and the proposed improvements to the text lead the reviewers to vote favorably for the paper.
>
> **DADS  baseline**
>
>
> | Method | MI-metric |
> |--------|----------|
> | Ant full method  | 1.33 ± 0.11    |
> | Ant DADS full | 0.33 ± 0.06    |
> | Humanoid full method    |1.29 ± 0.25    |
> | Humanoid DADS full  | 0.23 ± 0.05 |
>
> We ran a DADS baseline with the hyperparameters tuned by the original authors of the DADS paper. We ran seven seeds in each environment and will run three more seeds to match the ten seeds of our experiment. In the Humanoid environment the DADS agent does not learn to survive at all, i.e. it never reaches more than 200 reward in the original task, compared to the 9k of our approach. We will add this Baseline to the paper.
>
> **Random Network distillation baseline**
>
> | Method | MI-metric | zeroshot transfer |
> |--------|----------|----------|
> | Ant full method  | 1.33 ± 0.11    | 2506 ± 511     |
> | Ant RND | 0.27 ± 0.15    | 50 ± 189  |
> | Humanoid full method    |1.29 ± 0.25    | 9092 ± 1063     |
> | Humanoid RND    | 0.94 ± 0.17 | 7734 ± 1977 |
>
>
> We implemented Random Network Distillation (Burda et al. 2018) for the robotic environments. We ran a hyperparameter search for the RND specific hyperparameters with optuna. We searched for 100 respectively 60 trials of 200M timesteps optimizing the MI metric of the saved checkpoints in the Ant respectively Humanoid environment. We train both the Agent and the Prediction networks online and save a checkpoint every 20M timesteps. This is the same number of steps as we use for each skill.
>
> Then, we ran the best hyperparameters for 6 (Ant) and 10 (Humanoid) random seeds until they created 50 skills each. In the above table we report the result and refer to Figure 12 and 13 in Appendix J where we added plots in the style of Figure 6 for each of the seeds. We will run four additional seeds in the Ant environment to match the ten seeds used in all other experiments.
>
> We can see that our method outperforms RND heavily on the Ant environment. In the Humanoid environment the difference is smaller, but still noticeable. When looking at the plots in Appendix J we can also see quite a noticable difference between the two environments. In the Ant environment, the states visited by the policy overlap almost completely and each skill covers a large region of the state space. In the Humanoid, they still overlap and seem to be broader than the ones for our method (cf. Figure 9), but the difference is less striking.
>
> As we already discussed in our first response, the goal of methods like RND and ICM (Pathak et al. 2017) is to give rewards in all novel states. At any point in time the reward function rewards many different behaviors. This leads to broader skills compared to our method which only rewards a specific region of the state space in each iteration. We believe that these RND experiments highlight this difference between our method and the intrinsic motivation literature. In the Humanoid environment, there is a greater pressure from the environment for deterministic skills. Skills with higher entropy result in less stable gaits. Thus the skills are forced to be more narrow by the environment. In contrast, the robot in the Ant environment is more stable and thus this pressure is not as strong. We believe that this difference between the two environments results in the significant difference in empirical performance and the qualitative evaluation of the skills.
>
>
> **Ablations for the robotic environments**
>
>
> | Method | MI-metric | zeroshot transfer |
> |--------|----------|----------|
> | Ant full method  | 1.33 ± 0.11    | 2506 ± 511     |
> | Ant policy ablation | 1.09 ± 0.16    | 1731 ± 634     |
> | Ant value ablation  | 1.28 ± 0.15    | 2246 ± 794     |
> | Humanoid full method    |1.29 ± 0.25    | 9092 ± 1063     |
> | Humanoid policy ablation    |1.01 ± 0.09 | 8906 ± 616|
> | Humanoid value ablation    |0.88 ± 0.19 | 7356 ± 816 |
>
> In this table we report the full results of the ablation experiment discussed in the first response with five additional seeds to match the ten seeds of the original method.

---

> > ### Author Response · Authors · 2022-08-09
> > **Thanks to all the reviewers**
> >
> > We appreciate that reviewers have thoroughly gone over the paper, our responses and gave very useful feedback. We'll continue working hard to add another Intrinsic Motivation baseline with checkpointing for the final version of the paper. On top of improving our baselines, we believe this builds new connections between the skill discovery literature and the Intrinsic Motivation one.

---

### Meta-Review · Area_Chair_4So5 · 2022-08-25

**Recommendation:** Accept
**Confidence:** Certain

**Metareview:**

After a strong rebuttal from the authors and an extensive discussion among the reviewers, I believe the paper's pros outweigh its cons and this paper will be a valuable contribution to NeurIPS. I recommend it for acceptance and encourage the authors to address the reviewers comments for the camera-ready version of the paper, especially regarding the newly added baselines and other comparisons to SOTA approaches listed by the reviewers.


**Award:**

No

---

### Decision · Program_Chairs · 2022-09-14

Accept